# Temporal Changes in the Oxyhemoglobin Dissociation Curve of Critically Ill COVID-19 Patients

**DOI:** 10.3390/jcm11030788

**Published:** 2022-01-31

**Authors:** Samuele Ceruti, Bruno Minotti, Andrea Glotta, Maira Biggiogero, Giovanni Bona, Martino Marzano, Pietro Greco, Marco Spagnoletti, Christian Garzoni, Karim Bendjelid

**Affiliations:** 1Department of Critical Care, Clinica Luganese Moncucco, 6900 Lugano, Switzerland; andrea.glotta@moncucco.ch; 2Emergency Department, St. Gallen Cantonal Hospital, 9000 St. Gallen, Switzerland; bruno.minotti@kssg.ch; 3Clinical Research Unit, Clinica Luganese Moncucco, 6900 Lugano, Switzerland; maira.biggiogero@moncucco.ch (M.B.); giovanni.bona@outlook.com (G.B.); 4Department of Internal Medicine, Clinica Luganese Moncucco, 6900 Lugano, Switzerland; martino.marzano@gmail.com (M.M.); christian.garzoni@moncucco.ch (C.G.); 5Emergency Department, Clinica Luganese Moncucco, 6900 Lugano, Switzerland; pietro.greco@moncucco.ch (P.G.); marco.spagnoletti@moncucco.ch (M.S.); 6Intensive Care Division, Geneva University Hospitals, 1205 Geneva, Switzerland; karim.bendjelid@hcuge.ch

**Keywords:** COVID-19, dyspnea etiology, hypoxemia, hypoxia etiology, oxyhemoglobin metabolism, viral pneumonia, SARS-CoV-2

## Abstract

Critical COVID-19 is a life-threatening disease characterized by severe hypoxemia with complex pathophysiological mechanisms that are not yet completely understood. A pathological shift in the oxyhemoglobin curve (ODC) was previously described through the analysis of p50, intended as the oxygen tension at which hemoglobin is saturated by oxygen at 50%. The aim of this study was to analyze Hb-O_2_ affinity features over time in a cohort of critically ill COVID-19 patients, through the analysis of ODC p50 behavior. A retrospective analysis was performed; through multiple arterial blood gas (ABG) analyses, each p50 was calculated and normalized according to PaCO_2_, pH and temperature; patients’ p50 evolution over time was reported, comparing the first 3 days (early p50s) with the last 3 days (late p50s) of ICU stay. A total of 3514 ABG analyses of 32 consecutive patients were analyzed. The majority of patients presented a left shift over time (*p* = 0.03). A difference between early p50s and late p50s was found (20.63 ± 2.1 vs. 18.68 ± 3.3 mmHg, *p* = 0.03); median p50 of deceased patients showed more right shifts than those of alive patients (24.1 vs. 18.45 mmHg, *p* = 0.01). One-way ANOVA revealed a p50 variance greater in the early p50s (σ^2^ = 8.6) than in the late p50s (σ^2^ = 3.84), associated with a reduction over time (*p* < 0.001). Comparing the Hb-O_2_ affinity in critically ill COVID-19 patients between ICU admission and ICU discharge, a temporal shift in the ODC was observed.

## 1. Introduction

The oxyhemoglobin dissociation curve (ODC) is a sigmoid curve representing the relationship between arterial hemoglobin saturations (SaO_2_) at different arterial blood oxygen tensions (PaO_2_) [1,2]; the continuous correlation between SaO_2_ and PaO_2_ is reported as hemoglobin–oxygen (Hb-O_2_) affinity. ODC was first described by Christian Bohr in 1904 [3], who demonstrated how this affinity was mathematically represented by a sigmoid function.

p50 is the oxygen tension at which hemoglobin is saturated at 50%; in normal conditions, it ranges from 24 to 28 mmHg [4]. It follows the fact that a specific p50 value corresponds to a specific ODC curve and, therefore, to a specific Hb-O_2_ affinity. Various biochemical variables can physiologically influence the Hb-O_2_ affinity, leading to an ODC shift and to a consequent p50 change [5], with right and left shifts occurring in relation to changes in blood pH, PaCO_2_, temperature and 2,3-biphosphoglycerate (2,3-BPG) plasma concentration (Bohr/Haldane effect). The ODC shift can also be induced by other physiological or pathological conditions, such as aging [6], diabetes mellitus [7], anesthesia [8] and respiratory failure [9]. Considering these variables, an equation to represent the corrected ODC and its p50 was developed. In 1910, Archibald Vivian Hill, a British physiologist, was the first to elaborate on this equation [10], which was then revised by John Severinghaus in 1979 [11] and subsequently by many authors. Dash et al. [12] proposed a simple accurate mathematical model based on the original Hill formula with variation for O_2_, CO_2_, pH, 2,3-BPG and temperature (Appendix A).

COVID-19 is a disease characterized by severe hypoxemia [13] with complex pathophysiological mechanisms that are not yet completely understood [14]. In the acute phase, tachypnea and hypoxemia usually develop; quite surprisingly, and in contrast with similar conditions affecting the respiratory physiology, severe hypoxemia in COVID-19 is not always associated with dyspnea [14,15,16,17]. Some authors hypothesized that this discrepancy may be due to ODC shifting in the context of SARS-CoV-2 infection, possibly due to direct viral modulation on Hb structure, as evidenced by Wenzhong et al. [18] and Lanxioux et al. [19]. Vogel et al. [20] performed a retrospective analysis on arterial blood gas (ABG) analyses of more than 3500 blood gas analyses in 43 critically ill COVID-19 patients, comparing them with historical controls of patients with acute hypoxemia but without SARS-CoV-2 infection, observing a left shift of the ODC, whereas other authors did not find any changes [21,22]. Their results therefore suggest an in vivo increase in Hb-O_2_ affinity in critically ill COVID-19 patients. In contrast, Daniel et al. [21] observed that the in vitro Hb-O_2_ affinity in 14 patients affected by COVID-19 did not differ from that of 11 control patients in standard conditions; similarly, Pascual-Guardia et al. [22] did not find any change in the Hb-O_2_ affinity in COVID-19 patients.

Taking into consideration the temporal evolution of COVID-19 [23,24], and the feasible modification of the Hb structure induced by SARS-CoV-2 itself [18,19], we hypothesized temporal changes in the ODC shift in critically ill COVID-19 patients, which possibly correlates with pulmonary gas exchange, disease progression and survival. The aim of the study was to analyze Hb-O_2_ affinity features and their modifications over time through the analysis of normalized p50 behavior in a cohort of critically ill COVID-19 patients, hoping to aid in unveiling the underlying pathophysiological mechanism of this emerging disease in a pandemic scenario.

## 2. Materials and Methods

This was a retrospective observational study in a cohort of consecutive COVID-19 patients admitted from March to May 2020 to our COVID-19 Center in Lugano, Switzerland. The inclusion criteria were critically ill COVID-19 patients older than 18 years of age of both sexes undergoing invasive mechanical ventilation; diagnosis of COVID-19 was made by a positive RT-PCR molecular test. Patients excluded from the study were those with less than seven ABG analyses in the health record or with less than seven days in ICU care. Demographics and clinical and laboratory data were obtained from the hospital’s health medical record.

Each ABG analysis was carried out using one of the four ABL90 FLEX blood gas analyzers (Radiometer©) available in the intensive care unit (ICU) and emergency department (ED). Parameters such as the pH, PaO_2_, PaCO_2_, SaO_2_, sodium (Na^+^), potassium (K^+^), calcium (Ca^2+^), chloride (Cl^−^), glucose, L-lactate and hemoglobin were directly measured by the machine. Temperature and FiO_2_ at the time of arterial blood gas analysis were instead manually reported from the blood gas analyzer by the nurse in charge. Through specific queries already available in the health medical records, all patients’ ABG analyses, performed during the study period, were obtained, each of them reporting patients’ identification number and date and time of the analysis, as well as the pH, PaCO_2_, PaO_2_, SaO_2_, Na^+^, K^+^, Ca^2+^, Cl^−^, glucose, L-lactate and hemoglobin values. All the above-mentioned data were collected in an Excel dataset.

### 2.1. p50 Calculation with Variable Coefficient

We used the Dash formula [12] to calculate the ODC p50, starting from the PaO_2_/SaO_2_ relationship through the modified Hill coefficient according to Dash [10,12,25] and setting the parameters α, β and ɣ to 2.82, 1.2 and 29.25, respectively, in accordance with Severinghaus’ experimentally available data [11]. A weighted correction, balanced for PaCO_2_, pH and temperature, was implemented using the same Dash formula and the same definition as above, considering 2,3-BPG at 4.65 mmol/L as a constant (Appendix A) [1,4,20,26,27,28]. It was then possible to obtain a weighted p50 for each corresponding arterial blood gas analysis’ PaO_2_/SaO_2_ pairing, allowing a comprehensive analysis and comparison of the derived oxygen dissociation curves. According to the mathematical limitation in the Hill function [12], consisting of an excessive p50 increase when PaO_2_ reached high values (above 100 mmHg), we excluded all arterial blood gas analyses with PaO_2_ greater than 100 mmHg (Appendix A).

First, the standard ODC was calculated according to the original computations performed by Severinghaus et al. [11]. Based on the various modifications of patients’ p50 values occurring during the ICU stay, which were weighted according to the Hill and Dash formula [12], each corresponding ODC was then graphically reported and compared to the original one.

### 2.2. The p50 Shift over Time

For each patient, we described the p50 shift over time, plotting p50 values for each arterial blood gas analysis and thus representing the ODC temporal change. Moreover, to better characterize how relevant differences correlated with the disease severity evolution during the ICU stay, we calculated each patient’s p50s from the first three days in the ICU (early p50s) and compared them with those from the last three days in the ICU (late p50s). The time frame of three days was arbitrarily chosen as a balance between obtaining a high number of blood gas analyses and, at the same time, avoiding analyzing too many days of treatment altogether, especially given that numerous factors could rapidly alter the patients’ clinical condition (bacterial superinfection, sepsis, etc.).

Patients were further stratified according to their own temporal shift in “right-shift p50” and “left-shift p50”, and a further subgroup analysis was performed to identify any specific correlation, especially regarding p50 change over time. An additional comparison between all p50 in alive and deceased patients was performed. Finally, starting from the preliminary results, a large variability in the p50 of critically ill COVID-19 patients over time was identified. To better characterize this aspect and to define whether the p50 temporal evolution is constant and regular or fluctuates over time, a variance analysis for each single patient during all ICU hospitalizations was performed, which further analyzed any eventual variance in temporal progression.

### 2.3. Outcomes

The primary outcome was to assess the temporal p50 change in critically ill COVID-19 patients, comparing early p50s vs. late p50s, to describe the Hb-O_2_ affinity change over time. Secondary outcomes were to correlate the ODC and p50 variabilities with ICU survival and to stratify patients according to the temporal patterns of different p50 shifts, to analyze the p50 variability over time.

### 2.4. Statistical Analysis

Descriptive statistics were calculated; categorical data were reported as numbers (percentages). Numerical data distribution was reported as the mean (SD) or as median (IQR) according to statistical distribution, verified by the Kolmogorov–Smirnov test. Differences between continuous data were studied with the t-test for independent groups or with the Wilcoxon test if nonparametric analysis was required. The study of differences between the groups for categorical data was carried out with the chi-square test. A variance analysis was carried out with a one-way ANOVA through the clinical variables—sex, age, BMI, ICU length of stay (LOS), arterial hypertension, diabetes, chronic obstructive pulmonary disease (COPD) and obstructive sleep apnea syndrome (OSAS)—the prognostic variables—Simplified Acute Physiology Score (SAPS II), Sequential Organ Failure Assessment (SOFA) and the nine equivalents of nursing manpower use score (NEMS scores)—and laboratory variables—lactates, leucocytes, lymphocytes, thrombocytes, aspartate aminotransferase (AST), alanine aminotransferase (ALT), total bilirubin, C-Reactive Protein (CRP), lactate dehydrogenase (LDH), creatine kinase (CK), ferritin and creatinine; one-way multivariate analysis of variance (MANOVA) was also performed. All differences were reported as 2-tailed; an equal variance was not assumed. The confidence interval (CI) was 95%; the level of significance was established to be <0.05. No imputation for missing data was carried out. Statistical data analysis was performed using the SPSS 26.0 package (SPSS Inc., Chicago, IL, USA).

## 3. Results

During the enrollment period, 40 consecutive patients were admitted to the ICU, and 3747 total blood gas analyses were performed; 3 (7.5%) patients were excluded due to an ICU LOS less than 7 days, 5 (12.5%) patients were excluded due to a number of ABG analyses less than 7, 210 ABG analyses were excluded for a PaO_2_ greater than 100 mmHg. A total of 3514 ABG analyses from 32 patients were therefore analyzed (Figure 1).

A total of 29 (87.8%) patients were men, with a mean age of 62 years (SD 11.7); 12 (36.3%) patients were affected with diabetes, 14 (42.4%) with arterial hypertension, 3 (9%) with COPD, 5 (15.1%) with OSAS and 5 patients (15.1%) presented with a diagnosis of pulmonary embolism at ICU admission. The mean SAPS II score was 43 (SD 16), the median SOFA score was 6.5 (4.0–8.75) and the median NEMS score was 36 (19.25–39.0). All demographic, clinical and laboratory patient characteristics are reported in Table 1.

Regarding the arterial blood gas analyses, the mean pH was 7.39 (SD 0.08), with a mean PaCO_2_ of 48 mmHg (SD 12 mmHg), a mean PaO_2_ of 82 mmHg (SD 25 mmHg) and a mean bicarbonate level of 28 mmol/L (SD 5 mmol/L). All patients’ blood gas analyses results are reported in the Appendix A; the results of the blood gas analyses for each patient are reported in Table 2.

### 3.1. p50 Shift over Time

For each patient, a p50 calculation was reported, analyzing the mean (SD), median (IQR) and min/max values (Appendix A); for all patients, a Kolmogorov–Smirnov test rejected the hypothesis of normal data distribution. Analyzing the whole ICU population, a significant difference between early p50s and late p50s was found (20.63 ± 2.1 mmHg vs. 18.68 ± 3.3 mmHg, t = 2.15, df = 31, *p* = 0.03), identifying a left p50 shift over time (Figure 2 and Figure 3).

After stratifying patients according to ICU survival, we performed the same analysis; both groups presented a progressive left p50 shift over time. While the early p50s comparison between alive (*n* = 663 blood gas analyses) and deceased (*n* = 261 blood gas analyses) patients did not show any difference (t = −2.703, df = 30, *p* = 0.09), a significant difference was found comparing late p50s between the group of alive patients and the group of deceased patients. Indeed, the p50s of patients who survived were more left shifted than those who deceased (t = −3.29, df = 30, *p* = 0.01, Figure 4, Figure 5 and Figure 6). Moreover, matching the median of the p50 in alive compared with deceased patients, a significant difference was found (t = −4.08, df = 7.238, *p* = 0.004).

### 3.2. Subgroup Analysis

After analyzing the single patients’ temporal p50 shift, two different temporal patterns were identified (Appendix A). The 1st pattern showed a left p50 shift over time and was composed of 22 (68.7%) patients; conversely, the 2nd group presented a relative right p50 shift over time between the early and late phases and was composed of 10 (32.3%) patients. A chi-square analysis did not find any correlation between the pattern of p50 shift over time and any clinical factors, such as sex (*p* = 0.534), arterial hypertension (*p* = 0.267), diabetes (*p* = 0.438), OSAS (*p* = 0.637), COPD (*p* = 1.0), pulmonary embolism (*p* = 0.843), CVVHDF (*p* = 1.0), VAP (*p* = 1.0) and ICU survival (*p* = 0.632). Similarly, to analyze the effect of clinical, biological and laboratory exams on the temporal shift pattern of p50 and to investigate the specific contribution of these variables to the temporal ODC shift, multiple ANOVA (MANOVA) was performed. No significant differences between the covariance matrices concerning clinical parameters—such as the ICU survival (λ = 1.862, F(1;30) = 28.865, *p* = 0.183), BMI (λ = 0.015, F(1;27) = 23.576, *p* = 0.903), lactate (λ = 0.134, F(1;27) = 20.357, *p* = 0.717), scoring systems such as SOFA (λ = 2.376, F(1;27) = 26.980, *p* = 0.135), NEMS (λ = 2.741, F(1;27) = 25.374, *p* = 0.109), ICU LOS (λ = 0.048, F(1;30) = 26.167, *p* = 0.83)—or laboratory exams were revealed with the Levine test. Similarly, with the univariate results, no significant correlation was revealed (Appendix A).

### 3.3. Variance Analysis

During the ICU stay, p50 progression displayed several variations over time (Appendix A). Analyzing the variance of p50s for each patient, important temporal variability was confirmed, ranging from σ^2^ = 0.38 to σ^2^ = 31.86 (Appendix A). This variability did not result in constant and regular fluctuations during the entire ICU stay but instead showed important irregular fluctuations over time (Appendix A). A one-way ANOVA performed on each patient’s 3rd day groups of blood gas analyses confirmed a p50 temporal variability greater in the early p50s (σ^2^ = 8.6, 0.38–31.86) than in the late p50s intervals (σ^2^ = 3.84, 0.41–15.12); moreover, a p50 variance reduction over time was found (t = 4.198, df = 31, *p* < 0.001).

## 4. Discussion

Recent literature suggests that critically ill patients admitted for COVID-19 disease present a left shift in ODC, when compared with patients presenting with acute hypoxemia related to other diseases [20]. The present fact may contribute to understand the effects of this new disease on the Hb-O_2_ affinity.

In our retrospective analysis, we focused on p50 temporal shift in critically ill COVID-19 patients and found a significant difference in Hb-O_2_ affinity at ICU admission compared with the last three days of ICU stay. The identification of a disparity between early p50 and late p50 results suggests that one or more factors modulating Hb-O_2_ affinity are involved at these two time points. As the ODC was normalized according to pH, PaCO_2_ and temperature, it is relevant to consider that the difference in p50, which appears minimal in our finding, may be relevant from a pathophysiologic point of view. We can in fact affirm that, “despite” the normalization induced by the Hill formula, modified according to Dash, some additional—and thus far, undetermined—factor is able to induce a difference in p50s between ICU admission (early p50) and discharge (late p50).

As already stated by Vogel et al. [20], in our study, the first phase of the disease was characterized by a left shift in ODC, that goes beyond normal values associated with the Bohr/Haldane effect [5]. Therefore, although the ODC in COVID-19 patients is further shifted to the left compared with the standard condition, it showed an interesting tendency toward an absence of left shift in patients with negative outcomes, compared with alive patients. The present trend was not observed in those discharged from the ICU. In other words, a persistent left shift of the curve is observed in patients who will improve their lung function, while, compared with alive patients, an absence of left shift is observed, over time, in patients who will not show a recovery. We can consider, for deceased patients, the present result as a factual absence of p50 left shift trend, taking into account that it was calculated using the Hill coefficient, modified according to Dash (which permits us to avoid any confounding interactions). Based on previous studies, we can assume that, in the early phase, patients’ clinical status and lung injury are mainly determined by SARS-CoV-2 infection [29], while, in the late phase, lung inflammation’s impact on lungs and on the consequent gas exchange could be assumed to be dissimilar between patients [24,30].

Having assessed this ODC propensity to vary during the clinical course, a COVID-19-related effect (direct or indirect) on ODC appears possible. Some groups, such as Harutyunyan et al. [31] and Wenzhong et al. [18], have suggested that there is a hemoglobin allosteric modulation induced by SARS-CoV-2, possibly due to virally derived, transcribed nonstructural proteins (such as ORF) and surface proteins that can directly bind porphyrins and the heme group [18,32]. As a consequence, we observed a deviation of the ODC out of the normal range, even with the use of the Hill formula modified by Dash, which would translate into a possible change in the hemoglobin quaternary state [31,33]. In clinical practice, the mechanism involved in oxygen transport and release in critically ill COVID-19 patients remains to be deeply investigated [34,35]. However, we may speculate that an overall decreased left shift of the curve over time, in the history of the disease, is probably an indication of aggravation of the respiratory function and consequently of gas exchange. Contrarily, a relative steady left shift of the curve may indicate that the lung status is improving and therefore that tissues are relatively well oxygenated.

This study was burdened by some limits. First, we did not directly measure the viral load in the bloodstream; therefore, we can only hypothesize that it correlates with the p50 change. Studies comparing the viral load in the critical care setting and p50 are therefore needed to confirm this hypothesis; however, this is potentially supported by previous studies [24,30], which have confirmed a higher SARS-CoV-2 viral load at disease onset than at hospital discharge. Second, plasmatic 2,3-BPG measurements were not performed, thus leading to set the value at a default 4.65 mmol/L in all analyzed ABG analyses; although the 2,3-BPG level may have an impact on the p50 shift, in the acute setting, it would not seem to significantly influence the Hb-O_2_ affinity [36]. Third, the early and late ABG analyses, grouping in a 3-day interval was arbitrarily defined. The choice was a compromise between having an adequate number of ABG analyses, thus allowing a valid median p50 calculation, and the avoidance of combining too many days together, to be as precise as possible in describing the possible variability and the clinical course. Finally, ICU admission discharge times may not correspond to the same phase of illness for different patients. Although these data must be evaluated cautiously, they allow us to determine both the ODC and p50 shift over time, determining that this shift progressively reduces its fluctuation during the ICU stay.

## 5. Conclusions

An important variability and temporal left shift of the p50, ODC and Hb-O_2_ affinity in critically ill COVID-19 patients was observed. These data may bring forward an additional step toward understanding the pathophysiology of hypoxemia observed in COVID-19 patients. Interestingly, the absence of a steady left p50 shift over time may correlate with worse clinical outcomes in critically ill COVID 19 patients.

## Figures and Tables

**Figure 1 jcm-11-00788-f001:**
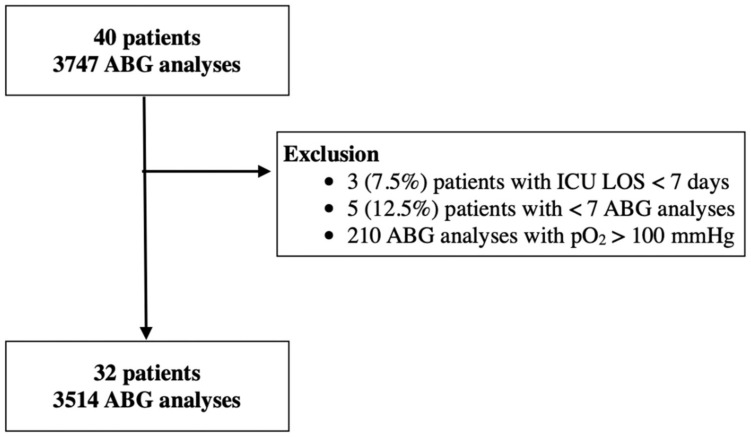
Patients flowchart: 40 consecutive critically ill COVID-19 patients were included (3747 ABG analyses). Patients with an ICU LOS inferior to 7 days and/or with a number of ABG analyses inferior to 7 were excluded, resulting in 32 patients included in the analysis, with a total of 3514 ABG analyses.

**Figure 2 jcm-11-00788-f002:**
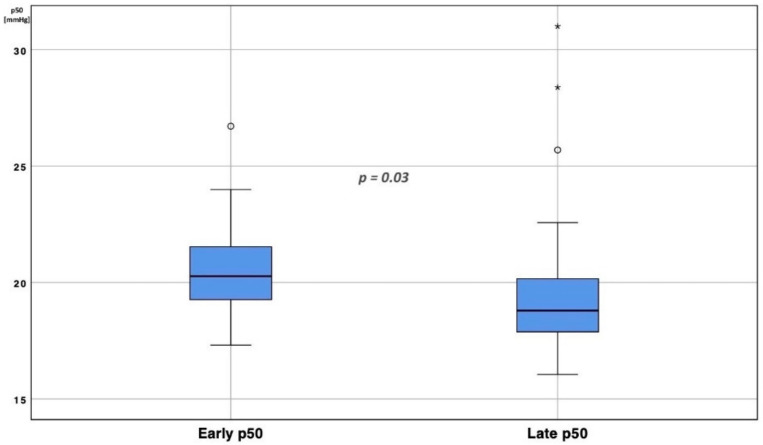
Early vs. late p50 comparison. Boxplot representation of p50s distribution in the first three days after ICU admission (early p50) compared with the last three days of ICU stay (late p50). Median early p50 was 20.12 mmHg (18.28–22.67); median late p50 resulted 19.09 mmHg (17.69–21.38). To eliminate all confounding factors of p50 modification, such as pH, PaCO_2_, 2,3-BPG and temperature, all p50s were calculated according to the Hill formula, modified by Dash. The circles represent outliers patients; * *p* < 0.01.

**Figure 3 jcm-11-00788-f003:**
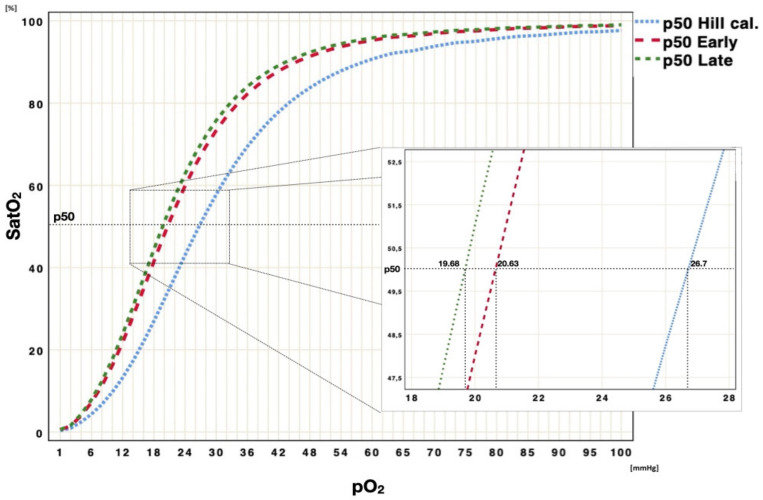
Graphic ODC p50 shift. Graphic representation of Hb-O_2_ affinity ODC, reporting the original Severinghaus curve (dashed blue curve), the ODC curve, identified by the median of early p50s (dashed red curve), and median of late p50s (dashed green curve). To eliminate all confounding factors on p50 modification, such as pH, PaCO_2_, 2,3-BPG and temperature, all p50s were calculated according to the Hill formula, modified by Dash.

**Figure 4 jcm-11-00788-f004:**
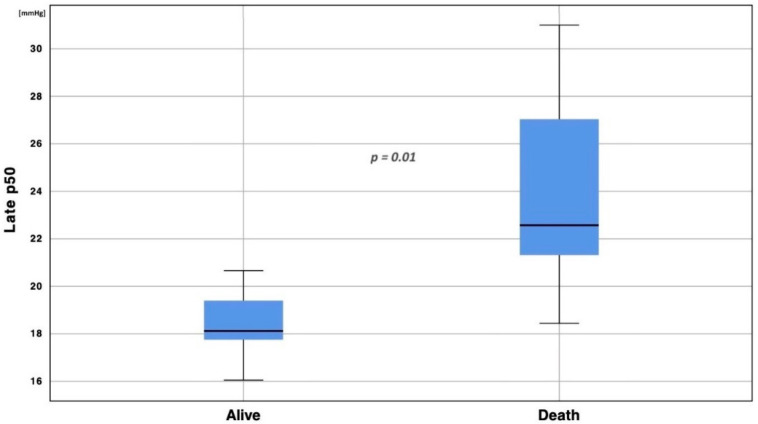
Late p50 stratification according to patients’ outcome. Boxplot analysis of late p50s between patients discharged from ICU alive and deceased (t = −3.29, df = 30, *p* = 0.01). To eliminate all confounding factors on p50 modification, such as pH, PaCO_2_, 2,3-BPG and temperature, all p50s were calculated according to the Hill formula, modified by Dash.

**Figure 5 jcm-11-00788-f005:**
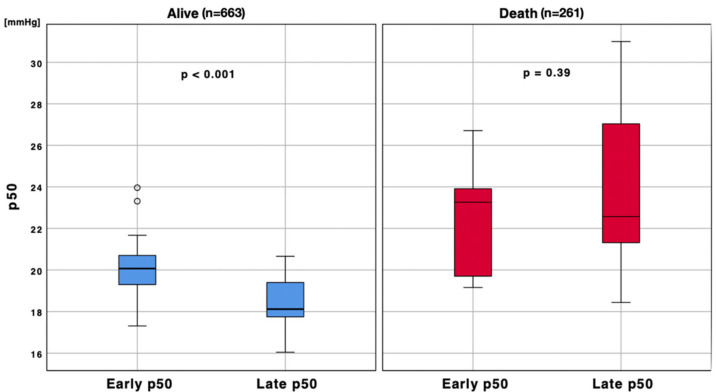
Early/late p50 stratification according to patients’ outcome. Boxplot analysis of early/late p50s between patients discharged from ICU alive (*n* = 663) and deceased (*n* = 261). To eliminate all confounding factors on p50 modification, such as pH, PaCO_2_, 2,3-BPG and temperature, all p50s were calculated according to the Hill formula, modified by Dash. The circles represent outliers patients.

**Figure 6 jcm-11-00788-f006:**
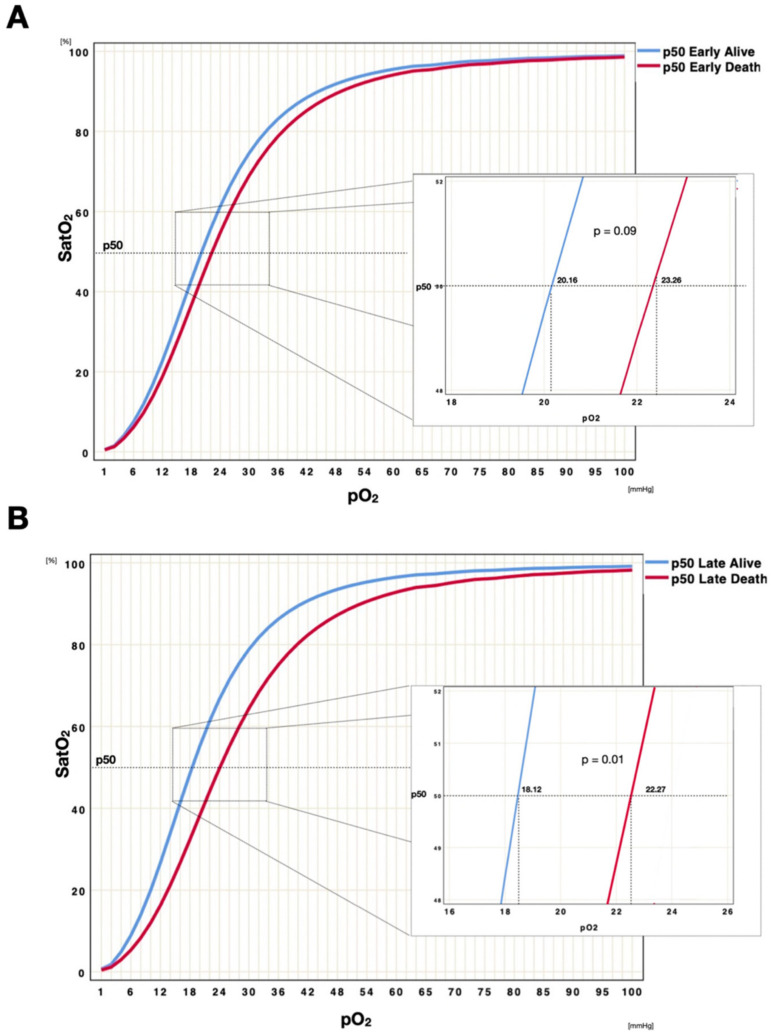
Early/late ODC p50 comparison according to patients outcome. Graphic representation of Hb-O_2_ affinity ODC, reporting the curve identified by the median of early p50s (**A**) and late p50s (**B**) of patients discharged alive (blue curve) and deceased (red curve). To eliminate all confounding factors on p50 modification, such as pH, PaCO_2_, 2,3-BPG and temperature, all p50s were calculated according to the Hill formula, modified by Dash. We may observe that patients with bad prognosis do not present a significant left shift over time.

**Table 1 jcm-11-00788-t001:** Patients’ clinical and laboratory characteristics.

Label	Unit	Mean (SD)/N (%)
Clinical characteristics		
Arterial Hypertension		14 (42.4)
Diabetes		12 (36.3)
OSAS		5 (15.1)
COPD		3 (9)
Pulmonary Embolism		5 (15.1)
CVVHDF		5 (15.1)
SAPS II		43 ± 6
SOFA		6.5 (4.0–8.75)
NEMS		36 (19.25–39.0)
LOS	days	14 (12–28)
ICU overall survival		25 (67.5)
Laboratory		
CRP	mg/L	245 ± 111
LDH	U/L	595 (482–770)
Total bilirubin	μmol/L	9.7 (7.6–16.4)
AST	U/L	52.5 (42.7–83.0)
ALT	U/L	42.0 (28.0–56.0)
Creatinine Kinase	U/L	337 (82–562)
White cells	G/L	7.7 (5.6–11.1)
Lymphocytes	G/L	0.8 (0.5–1.0)
Lactate	mmol/L	0.9 (0.8–1.5)
Ferritin	ng/mL	2101 (1046–3857)
Creatinine	μmol/L	88 (72–122)
Thrombocytes	G/L	242 ± 89
Arterial Blood gas analysis		
Number		3′514
pH		7.39 ± 0.08
PaCO_2_	mmHg	48 ± 12
PaO_2_	mmHg	82 ± 25
HCO_3_^−^	mmol/L	28 ± 5
BE	mmol/L	3 ± 4
SatO_2_	%	95 ± 15
Hemoglobin	g/dL	11.5 ± 2.0
Hematocrit	%	35 ± 7
COHb	%	1.3 ± 0.5
MetHb	%	0.9 ± 0.4
Na^+^	mmol/L	146 ± 6
K^+^	mmol/L	4 ± 0.8
Ca^2+^	mmol/L	1.14 ± 0.08
Cl^−^	mmol/L	110 ± 7
Glucose	mmol/L	8.9 ± 3.1
Temperature	°C	37.1 ± 0.4
Lactate	mmol/L	1.3 ± 0.5

CVVHDF—continuous veno-venous hemodiafiltration; BE—base excess; COHb—carboxy-hemoglobin; MetHb—methemoglobin.

**Table 2 jcm-11-00788-t002:** Patients’ ABG analysis data. Analyses was performed by ABL90 FLEX blood gas analyzer; parameters such as pH, PaO_2_, PaCO_2_, SaO_2_, sodium (Na^+^), potassium (K^+^), calcium (Ca^2+^), chloride (Cl^−^), glucose, lactate and hemoglobin were directly measured by the machine. Mean (SD) of patients’ relevant parameters was reported.

		Pt 1	Pt 2	Pt 3	Pt 4	Pt 5	Pt 6	Pt 7	Pt 8
* **n** *		168	21	79	236	88	66	75	46
pH		7.37 ± 0.04	7.44 ± 0.03	7.40 ± 0.05	7.41 ± 0.07	7.44 ± 0.05	7.44 ± 0.03	7.43 ± 0.04	7.44 ± 0.04
PaO_2_	[mmHg]	86.9 ± 20.1	87.3 ± 33.8	108.2 ± 43.9	85.3 ± 28.6	102.2 ± 29.5	92.9 ± 34.4	100.9 ± 32.5	112.2 ± 33.8
PaCO_2_	[mmHg]	43 ± 5.6	37.8 ± 4.9	42 ± 5.7	47.7 ± 8.9	44.7 ± 6.7	42 ± 4.4	36.7 ± 4.2	40.4 ± 6.4
HCO_3_	[mmol/L]	24.9 ± 2.0	25.2 ± 2.1	25.9 ± 3.4	29.9 ± 4.4	30 ± 3.3	28.7 ± 3.1	24.6 ± 3.7	27.2 ± 2.9
BE	[mmol/L]	−0.4 ± 1.8	1.2 ± 1.6	0.9 ± 3.3	4.5 ± 4.3	5.1 ± 3.1	4 ± 3.0	0.5 ± 3.7	2.7 ± 2.3
SaO_2_	[%]	95 ± 6	97 ± 2	97 ± 5	95 ± 7	97 ± 4	96 ± 2	97 ± 4	98 ± 2
Hb	[g/L]	9.7 ± 1.3	14.6 ± 1.6	13.2 ± 1.1	11.1 ± 2.8	12 ± 0.9	13.5 ± 2.7	11.4 ± 2.8	12.5 ± 3.1
Hct	[%]	30 ± 4	45 ± 5	40 ± 4	34 ± 9	37 ± 3	41 ± 8	35 ± 8	38 ± 9
O_2_Hb	[%]	93 ± 5	95 ± 2	95 ± 5	93 ± 4	95 ± 4	94 ± 3	95 ± 4	95 ± 2
COHb	[%]	1.6 ± 0.3	1.2 ± 0.3	1.0 ± 0.4	1.7 ± 0.4	1.3 ± 0.2	0.8 ± 0.3	0.7 ± 0.3	1.0 ± 0.4
MetHb	[%]	0.8 ± 0.4	0.4 ± 0.1	0.5 ± 0.2	0.6 ± 0.2	0.8 ± 0.3	1.1 ± 0.3	1.2 ± 0.3	1.1 ± 0.3
Na^+^	[mmol/L]	143 ± 5	141 ± 2	143 ± 3	152 ± 6	150 ± 4	148 ± 2	141 ± 4	147 ± 4
K^+^	[mmol/L]	4.4 ± 0.5	3.8 ± 0.2	3.9 ± 0.3	3.9 ± 0.4	4.2 ± 2.3	3.7 ± 0.3	4.0 ± 0.4	3.8 ± 0.4
Ca^2+^	[mmol/L]	1.09 ± 0.08	1.08 ± 0.04	1.17 ± 0.05	1.11 ± 0.07	1.16 ± 0.05	1.13 ± 0.05	1.11 ± 0.04	1.17 ± 0.06
Cl^−^	[mmol/L]	107 ± 5	109 ± 2	111 ± 4	114 ± 6	111 ± 5	111 ± 3	110 ± 4	113 ± 5
Glucose	[mmol/L]	9.7 ± 2.2	5.7 ± 1.3	7.3 ± 1.1	10.2 ± 4.1	9.5 ± 1.8	9.8 ± 1.9	6.2 ± 0.9	9.7 ± 1.8
Lactate	[mmol/L]	1.3 ± 0.3	0.8 ± 0.2	1.2 ± 0.2	1.2 ± 0.7	1.1 ± 0.3	1.0 ± 0.3	1.3 ± 0.4	1.1 ± 0.3
		Pt 9	Pt 10	Pt 11	Pt 12	Pt 13	Pt 14	Pt 15	Pt 16
* **n** *		155	75	164	103	77	56	125	204
pH		7.44 ± 0.08	7.42 ± 0.04	7.34 ± 0.07	7.38 ± 0.12	7.40 ± 0.06	7.43 ± 0.04	7.45 ± 0.04	7.40 ± 0.06
PaO_2_	[mmHg]	97.8 ± 37.7	89.8 ± 26.3	78.3 ± 24.8	83 ± 21.9	90.5 ± 20.2	85.4 ± 21.9	78.3 ± 14.8	72.8 ± 13.4
PaCO_2_	[mmHg]	46.4 ± 13.9	46.1 ± 6.9	53.3 ± 10.3	49 ± 17.1	42.4 ± 5.8	44.6 ± 4.1	41.5 ± 8.9	51.1 ± 10.3
HCO_3_	[mmol/L]	30.6 ± 4.1	29.8 ± 4.1	28.4 ± 3.5	27.3 ± 3.5	26.2 ± 3.2	29.3 ± 2.7	28.7 ± 5.2	30.8 ± 3.3
BE	[mmol/L]	5.6 ± 2.7	4.6 ± 3.6	1.9 ± 3.6	1.3 ± 3.8	1.1 ± 3.4	4.3 ± 2.8	4.2 ± 4.2	5.1 ± 2.8
SaO_2_	[%]	97 ± 2	96 ± 4	95 ± 5	96 ± 3	97 ± 2	96 ± 5	95 ± 2	94 ± 3
Hb	[g/L]	10.9 ± 2.1	12.4 ± 1.4	11.8 ± 2.0	13.7 ± 1.9	12.1 ± 1.6	12.4 ± 0.7	14.8 ± 1.0	10.5 ± 2.1
Hct	[%]	34 ± 6	38 ± 4	36 ± 6	42 ± 6	37 ± 5	38 ± 2	45 ± 3	32 ± 6
O_2_Hb	[%]	95 ± 3	94 ± 4	93 ± 5	94 ± 3	95 ± 2	94 ± 5	93 ± 2	91 ± 3
COHb	[%]	1.1 ± 0.4	0.6 ± 0.4	1.5 ± 0.2	1.5 ± 0.2	1.0 ± 0.4	1.3 ± 0.2	1.3 ± 0.3	1.8 ± 0.3
MetHb	[%]	1.2 ± 0.3	1.1 ± 0.3	0.6 ± 0.2	0.6 ± 0.2	1.0 ± 0.5	0.6 ± 0.2	1.2 ± 0.2	1.0 ± 0.3
Na^+^	[mmol/L]	152 ± 5	146 ± 5	147 ± 2	146 ± 4	148 ± 5	146 ± 3	153 ± 13	148 ± 3
K^+^	[mmol/L]	3.8 ± 2.2	3.6 ± 0.3	4.3 ± 0.6	4.0 ± 0.6	3.9 ± 0.4	3.8 ± 0.3	3.8 ± 0.4	4.1 ± 0.4
Ca^2+^	[mmol/L]	1.10 ± 0.04	1.13 ± 0.06	1.1 ± 0.1	1.12 ± 0.6	1.13 ± 0.1	1.16 ± 0.1	1.16 ± 0.1	1.14 ± 0.05
Cl^−^	[mmol/L]	113 ± 5	107 ± 5	111 ± 3	111 ± 5	103 ± 4	108 ± 3	117 ± 6	111 ± 3
Glucose	[mmol/L]	7.6 ± 1.4	6.9 ± 1.3	10.3 ± 2.6	9.0 ± 2.0	1.0 ± 0.3	11.1 ± 2.3	9.7 ± 2.5	8.9 ± 1.7
Lactate	[mmol/L]	1.2 ± 0.3	1.2 ± 0.4	2.1 ± 0.6	1.4 ± 0.5	1.0 ± 0.3	1.5 ± 0.5	1.5 ± 0.4	0.9 ± 0.3
		Pt 17	Pt 18	Pt 19	Pt 20	Pt 21	Pt 22	Pt 23	Pt 24
* **n** *		251	41	265	25	159	68	92	212
pH		7.40 ± 0.07	7.27 ± 0.07	7.38 ± 0.07	7.42 ± 0.03	7.41 ± 0.08	7.42 ± 0.06	7.41 ± 0.05	7.36 ± 0.07
PaO_2_	[mmHg]	75.4 ± 18.5	73.8 ± 19.9	79.1 ± 20.4	72.7 ± 18.8	83.7 ± 22.1	82.6 ± 24.0	74.7 ± 15.8	72.1 ± 16.0
PaCO_2_	[mmHg]	40.7 ± 7.9	52.0 ± 11.0	59.4 ± 15.8	37.7 ± 2.7	46.6 ± 11.9	43.7 ± 8.6	54.2 ± 9.0	55.4 ± 9.0
HCO_3_	[mmol/L]	24.7 ± 3.0	23.2 ± 2.8	34.9 ± 8.5	24.7 ± 2.4	29.0 ± 4.5	28.2 ± 3.9	34.1 ± 4.2	30.9 ± 4.4
BE	[mmol/L]	−0.1 ± 3.0	−4.2 ± 2.1	8.5 ± 7.8	0.4 ± 2.5	3.8 ± 3.3	3.2 ± 3.3	7.7 ± 3.5	4.5 ± 4.4
SaO_2_	[%]	94 ± 4	92 ± 6	94 ± 4	94 ± 4	102 ± 70	96 ± 4	94 ± 5	94 ± 4
Hb	[g/L]	8.8 ± 1.2	14.5 ± 1.0	9.5 ± 1.8	14.1 ± 0.8	11.9 ± 1.1	13.9 ± 0.9	13.5 ± 0.9	9.7 ± 1.8
Hct	[%]	27 ± 4	44 ± 3	29 ± 5	43 ± 2	37 ± 3	43 ± 3	41 ± 3	30 ± 6
O_2_Hb	[%]	92 ± 4	90 ± 5	92 ± 4	92 ± 4	94 ± 2	94 ± 3	92 ± 5	92 ± 3
COHb	[%]	1.4 ± 0.6	1.1 ± 0.1	1.7 ± 0.6	1.1 ± 0.1	1.2 ± 0.2	1.3 ± 0.2	1.1 ± 0.2	1.5 ± 0.3
MetHb	[%]	0.9 ± 0.3	0.9 ± 0.3	1.0 ± 0.4	0.8 ± 0.3	1.0 ± 0.2	1.0 ± 0.2	1.0 ± 0.2	1.0 ± 0.3
Na^+^	[mmol/L]	141 ± 4	145 ± 3	149 ± 4	144 ± 2	149 ± 6	146 ± 5	147 ± 4	146 ± 5
K^+^	[mmol/L]	4.4 ± 0.6	4.4 ± 0.6	3.8 ± 0.4	3.6 ± 0.3	4.0 ± 0.4	3.8 ± 0.4	4.1 ± 0.4	3.9 ± 0.4
Ca^2+^	[mmol/L]	1.16 ± 0.1	1.11 ± 0.1	1.15 ± 0.08	1.13 ± 0.03	1.16 ± 0.08	1.11 ± 0.05	1.20 ± 0.06	1.21 ± 0.04
Cl^−^	[mmol/L]	107 ± 4	115 ± 4	108 ± 10	110 ± 2.8	113 ± 5	109 ± 13	104 ± 3	109 ± 6
Glucose	[mmol/L]	9.6 ± 3.7	9.1 ± 1.2	8.7 ± 1.7	5.7 ± 0.6	8.4 ± 2.5	6.0 ± 1.1	8.9 ± 2.0	7.7 ± 1.8
Lactate	[mmol/L]	1.2 ± 0.4	2.1 ± 0.6	1.4 ± 0.8	1.2 ± 0.3	1.3 ± 0.5	1.3 ± 0.4	1.7 ± 0.5	1.1 ± 0.3
		Pt 25	Pt 26	Pt 27	Pt 28	Pt 29	Pt 30	Pt 31	Pt 32
* **n** *		41	130	14	224	54	38	36	126
pH		7.40 ± 0.06	7.39 ± 0.09	7.40 ± 0.04	7.34 ± 0.06	7.44 ± 0.03	7.45 ± 0.04	7.46 ± 0.03	7.31 ± 0.11
PaO_2_	[mmHg]	96.7 ± 29.0	80.8 ± 19.6	65.4 ± 14.0	78.6 ± 18.7	80.2 ± 24.9	85.7 ± 19.6	61.0 ± 10.4	71.4 ± 14.0
PaCO_2_	[mmHg]	40.6 ± 4.9	54.8 ± 13.3	37.2 ± 3.8	49.1 ± 7.4	41.3 ± 5.0	41.7 ± 3.7	37.9 ± 6.5	60.8 ± 19.1
HCO_3_	[mmol/L]	25.2 ± 2.4	32.9 ± 5.8	23.2 ± 3.6	26.5 ± 2.4	27.7 ± 2.9	28.8 ± 2.5	27.7 ± 1.3	29.6 ± 3.4
BE	[mmol/L]	0.4 ± 3.0	6.5 ± 5.4	−1.4 ± 3.7	0.3 ± 2.4	3.1 ± 2.5	4.3 ± 2.6	3.7 ± 1.3	2.5 ± 2.9
SaO_2_	[%]	97 ± 3	95 ± 3	91 ± 9	95 ± 3	95 ± 2	94 ± 14	91 ± 4	93 ± 4
Hb	[g/L]	14.7 ± 1.0	13.5 ± 1.1	10.7 ± 1.3	10.6 ± 2.1	12.2 ± 0.5	13.3 ± 1.5	14.5 ± 1.0	11.4 ± 1.8
Hct	[%]	45 ± 3	41 ± 3	33 ± 4	33 ± 7	37 ± 2	41 ± 5	44 ± 3	35 ± 5
O_2_Hb	[%]	95 ± 3	93 ± 3	89 ± 8	93 ± 3	93 ± 3	94 ± 4	90 ± 4	90 ± 4
COHb	[%]	1.08 ± 0.1	1.4 ± 0.2	1.26 ± 0.1	1.7 ± 0.3	1.0 ± 0.1	1.0 ± 0.2	1.02 ± 0.1	1.43 ± 0.3
MetHb	[%]	0.9 ± 0.4	1.1 ± 0.3	0.7 ± 0.2	0.7 ± 0.3	1.2 ± 0.3	1.2 ± 0.3	0.9 ± 0.4	1.1 ± 0.3
Na^+^	[mmol/L]	146 ± 3	145 ± 4	138 ± 5	148 ± 5	144 ± 15	142 ± 3	141 ± 1	145 ± 3
K^+^	[mmol/L]	3.8 ± 0.2	4.0 ± 0.5	3.8 ± 0.2	4.4 ± 0.5	3.8 ± 0.4	3.8 ± 0.3	3.6 ± 0.3	4.5 ± 1.0
Ca^2+^	[mmol/L]	1.12 ± 0.04	1.13 ± 0.04	0.98 ± 0.37	1.19 ± 0.06	1.11 ± 0.07	1.10 ± 0.04	1.11 ± 0.04	1.18 ± 0.04
Cl^−^	[mmol/L]	112 ± 4	107 ± 5	104 ± 4	111 ± 8	108 ± 5	106 ± 4	106 ± 2	109 ± 10
Glucose	[mmol/L]	6.2 ± 0.8	8.8 ± 1.5	6.3 ± 2.8	10.1 ± 7.3	10.3 ± 3.6	7.5 ± 1.1	6.1 ± 1.2	8.0 ± 1.5
Lactate	[mmol/L]	1.6 ± 0.4	1.2 ± 0.3	0.6 ± 0.2	1.3 ± 0.4	1.7 ± 0.8	1.1 ± 0.2	1.6 ± 0.3	1.2 ± 0.4

## Data Availability

The data presented in this study are available on request from the corresponding author. The data are not publicly available due to privacy and ethical restrictions.

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
