# Peer review of "Temporal Changes in the Oxyhemoglobin Dissociation Curve of Critically Ill COVID-19 Patients"

_jcm, 2022, doi:10.3390/jcm11030788_

Round 1
Reviewer 1 Report
Thank you for the modified manuscript.
I have 2 minor comments:
Abstract : twice the same sentence at the end
Page 2 line 83: choose between observed and demonstrated
Author Response
We thank the reviewer for these suggestions. We’ve modified the text according to these indications, removing the duplicate sentence at the end of the abstract and removing “demonstrated” that was a typo from a previous revision. Moreover, we performed a minor spell check as suggested.

Reviewer 2 Report
After reviewing the current version of the manuscript, in which the authors have taken into account the comments and suggestions of previous reviewers.
Author Response
We thank the reviewer to appreciate our revision, which has been performed in agreement with the received suggestions.

This manuscript is a resubmission of an earlier submission. The following is a list of the peer review reports and author responses from that submission.
Round 1
Reviewer 1 Report
Ceruti et al described temporal evolution of p50 ODC in critically ill and ventilated COVID patients, comparing early (3 first days) and late (3 last days) p50.
I don’t have access to the supplementary materials?, please provide them, I cannot conclude my review before to see them.
Abstract:
explain what p50 is
Correct “p50 evolution over time was reported” instead of “were”
Typology: suppress italic in the abstract
“Temporal evolution …. supports the role of SARS-CoV-2 as an effector involved in the allosteric modulation…”: can we really conclude that with the results of this study?
Key words: suppress dyspnea blood (dyspnea is a subjective clinical sign) and hypoxia blood (hypoxemia?)
Viral pneumonia instead of pneumonia viral
The manuscript requires editing by a native English speaker
Introduction:
Page 2 line 67 and 71: Homogenize references into brackets ( ) or [ ]
Material and methods:
Page 3 line 101 102: suppress “and analyzed which were obtained”
Page 3 line 105: explain abbreviation when used for the first time in text: ICU Intensive Care Unit and ED Emergency Department
Page 4 line 180: replace “missing data was not included” by “no imputation for missing data was carried out”
Page 4 line 173: SAPS: is it SAPS II? If it is please correct it in the whole manuscript
Results
Table 1: suppress results that are redundant with text (demographics and clinical characteristics)
Page 7 line 215 : correct typo of carbox-yhemoglobin (instead of yhe-moglobin)
I don’t see the interest of a mean of arterial blood gases over the hospitalization, among ventilated patients with a LOS of 14 days? They could vary a lot during the stay?
Conclusion
P 14 line 394, 395: The authors should stay factual: the results of this study do not support or demonstrate the role of SARS Cov2. Please only describe your own observation (shift).
Author Response
I don’t have access to the supplementary materials? please provide them, I cannot conclude my review before to see them.
We proceeded to submit the SM with the R1 documents.
Abstract
explain what p50 is
Correct “p50 evolution over time was reported” instead of “were”
Typology: suppress italic in the abstract
“Temporal evolution …. supports the role of SARS-CoV-2 as an effector involved in the allosteric modulation…”: can we really conclude that with the results of this study?
Key words: suppress dyspnea blood (dyspnea is a subjective clinical sign) and hypoxia blood (hypoxemia?)
Viral pneumonia instead of pneumonia viral
The manuscript requires editing by a native English speaker
Response:
We thank the reviewer for these suggestions. We proceeded to perform the suggested changes in the Abstract (lines 26-27, line 32), as well as to totally remove the sentences indicated, remaining more factual concerning the conclusion. In addition to the English Editing Certificate attached, we implemented the revision with an editing by a native English speaker.
Introduction
Page 2 line 67 and 71: Homogenize references into brackets ( ) or [ ]
Response:
We thank the reviewer for this indication. We proceeded perform the suggested changes.
Material and methods
Page 3 line 101 102: suppress “and analyzed which were obtained”
Page 3 line 105: explain abbreviation when used for the first time in text: ICU Intensive Care Unit and ED Emergency Department
Page 4 line 180: replace “missing data was not included” by “no imputation for missing data was carried out”
Page 4 line 173: SAPS: is it SAPS II? If it is please correct it in the whole manuscript
Response:
We thank the reviewer for these suggestions. We proceeded to perform the suggested changes, as well as to modify all the indicated sentences.
Results
Table 1: suppress results that are redundant with text (demographics and clinical characteristics)
Page 7 line 215 : correct typo of carbox-yhemoglobin (instead of yhe-moglobin)
I don’t see the interest of a mean of arterial blood gases over the hospitalization, among ventilated patients with a LOS of 14 days? They could vary a lot during the stay?
Response:
We thank the reviewer for these suggestions, we proceeded to modify the text and table 1 as suggested; moreover, we appreciate the opportunity to clarify the second most important aspect of the manuscript. It is relevant to report that the number of blood gas analyses analyzed during the study was relevant, both for a single patient than for all included patients. In critical moments during the ICU stay, such as the ICU admission or during the onset of ventilation-associated pneumonia, acute kidney injury, bloodstream infections, etc, patients could present theoretically relevant changes in their own blood gas analyses. The analysis of the mean values allowed us to show that the oxygenation (pO2 and SaO2) was the only altered data in the blood gas analysis, while the acid-base balance and the rest of the metabolic picture remained substantially unchanged. Furthermore, pCO2, pH and temperature could have influenced the p50 due to the Bohr effect/Haldane effect; in addition to the Hill and Dash analysis we reported, to weight the p50 alteration on the basis of "confounding factors", we considered appropriate to show the mean values ​​analyzed during the entire ICU stay, to underline that no differences from normal values were found, apart from hypoxemia.
Conclusion
P 14 line 394, 395: The authors should stay factual: the results of this study do not support or demonstrate the role of SARS Cov2. Please only describe your own observation (shift).
Response:
We thank the reviewer for this important suggestion. According to this indication, we removed any link related to SARS-COV-2 role in Hb-O2 affinity shift, remaining more factual and reporting only our observations (lines 336-344); the same modification was also reported in the conclusion of the Abstract (lines 40-43).

Reviewer 2 Report
Thank you for the opportunity to review this manuscript.
The authors present an exploratory retrospective analysis on blood gas data from COVID-19 patients regarding hemoglobin affinity.
Major comments:
First, I think more sophisticated analysis methods are needed to properly interpret the data. The data are many repeated measures within subjects. The current analysis does not take into account that the data are repeated measures within a subject. The authors could use analysis methods such as linear mixed models (continuous outcome) or generalized estimating equations (categorical data) to account for this.
Second, the current conclusions (and title) are not supported by the currently presented results (even though I think different methods need to be used). The title suggests that changes in affinity are related to mortality. What the authors actually found was that the late p50 was higher in patients who died, but there was no association between temporal changes and outcome.
Third, the discussion of the results reads somewhat teleological. A left change is presented as an ‘adaptive’ mechanism that the human body somehow is able to turn on, and it turns it on because it is good for itself. I think it would be more scientifically sound to just discuss whatever is observed in patients and how this may be caused. The teleological argument that the authors present is also difficult to match with the actual observations. The authors seem to want to eat one’s cake and have it too. If a left shift is induced by SARS-COV2 or the body’s response to the virus, would you expect left shift to be maximal when patients have higher viral loads and are sicker (at ICU admission vs. discharge)? The presented data seem to indicate that left shift is associated with less viral load.
Minor comments:
Analyses is the plural of analysis.
The outcome paragraph is currently written as an aims statement.
The sentence in line 316 suggests that the change in p50 was from an analysis in only survivors, while that result seems to have been obtained in both alive and dead patients?
Line 399 sounds promotional.
Author Response
- First, I think more sophisticated analysis methods are needed to properly interpret the data. The data are many repeated measures within subjects. The current analysis does not take into account that the data are repeated measures within a subject. The authors could use analysis methods such as linear mixed models (continuous outcome) or generalized estimating equations (categorical data) to account for this.
Response:
We thank the reviewer for this interesting suggestion. The use of linear mixed models for continuous outcome could represent the most appropriate method to better study the temporal evolution of p50, since these are repeated measurements over time of the same subjects. We took into consideration this analysis at the beginning of our study, but we realized that it was not feasible. Critically ill COVID-19 patients presented an ICU length-of-stay (LOS) relatively long and burdened of complications, such as pulmonary superinfections, severe acute kidney injury requiring CVVHDF, septic shocks from bloodstream infection, etc... which make it difficult to obtain "clean" and bias-free data to get an uncontaminated analysis of the p50 temporal changes. Based on the pathophysiological evolution of the critically ill COVID-19 disease, and confirmed by Luyt CE et al [Ann Intensive Care 2020], Rouzé et al [Intensive Care Medicine 2021] and Torres A et al [Eur Respir J 2021] and in the ESICM guidelines about critically ill COVID-19 management, the rate of superinfection and further complications for the first days of hospitalization in the ICU resulted almost zero. These reasons induced us to analyze and compare the dissociation of hemoglobin at the ICU admission with the last 3 days of the ICU stay.
Due to the need to obtain an adequate functional muscular and respiratory recovery in order to avoid early ICU re-admission (< 72h), the relative long ICU hospitalization allowed to obtain an infectious-inflammatory condition better than at the ICU admission. We cannot evaluate the viral load at the ICU discharge, but according to To KK et al [Lancet Infect Dis 2020] patients with COVID-19 presented higher viral load at the onset of the disease compared to ICU discharge; our clinical and biological data confirm us that the infectious-inflammatory condition resulted definitely improved, if compared to the ICU admission. For this reason, we performed a comparison between these two specific periods of the ICU hospitalization, which are certainly free of biases and allow to obtain a clean and reproducible result, with an apparently better continuous analysis of the p50 shift.
However, we understand the indication of the reviewer and we proceeded to remove the word evolution in the manuscript, also in the primary outcome, with a more adequate terms such as change or shift that best represent the statistical analysis we have performed (lines 143, line 153, 234, line 239).
Second, the current conclusions (and title) are not supported by the currently presented results (even though I think different methods need to be used). The title suggests that changes in affinity are related to mortality. What the authors actually found was that the late p50 was higher in patients who died, but there was no association between temporal changes and outcome.
Response:
We thank the reviewer for this important aspect. According to this indication, we modified the manuscript removing all possible connection between p50 temporal shift and any clinical outcome, remaining more factual and adhering to our data. We changed the title, removing all connection with clinical outcome. The Abstract and the Discussion were modified removing all sentences suggesting any association between temporal changes and outcome; Discussion and the Conclusions have been improved and we were more cautious, remaining more factual and descriptive (lines 40-43, lines 380-383).
- Third, the discussion of the results reads somewhat teleological. A left change is presented as an ‘adaptive’ mechanism that the human body somehow is able to turn on, and it turns it on because it is good for itself. I think it would be more scientifically sound to just discuss whatever is observed in patients and how this may be caused. The teleological argument that the authors present is also difficult to match with the actual observations. The authors seem to want to eat one’s cake and have it too. If a left shift is induced by SARS-COV2 or the body’s response to the virus, would you expect left shift to be maximal when patients have higher viral loads and are sicker (at ICU admission vs. discharge)? The presented data seem to indicate that left shift is associated with less viral load.
Response:
We appreciate the opportunity to clarify this important aspect of the manuscript. We proceeded to modify the Discussion as indicated by the reviewer, comparing the changes found in the p50 evolution curves with the well-known Bohr effect / Haldane effect. We also removed two sentences with potential teleological arguments, as indicated by the reviewer, remaining more descriptive about the p50 shift.
All ODC curves of critically ill COVID-19 patients presented a left shift compared to normal p50 values (24-28 mmHg), especially in the early stage of the disease. In patients who died at the end of ICU stay, the change was relatively more shifted to the right if compared to survivors, but still remaining to the left if compared to normal p50 values. To better clarify this relevant aspect, we integrated and explained in the Discussion the term “absence of left shift” referred to deceased patients compared to alive patients (line 342), in order to avoid possible misunderstanding as identified by the reviewer.
- Minor comments:
Analyses is the plural of analysis.
The outcome paragraph is currently written as an aims statement.
The sentence in line 316 suggests that the change in p50 was from an analysis in only survivors, while that result seems to have been obtained in both alive and dead patients?
Line 399 sounds promotional.
Response:
We thank the reviewer for these suggestions. We have proceeded to carry out the proposed changes regarding grammar errors, as well as to modify the sentences indicated: the sentence in line 252-260 referred to both living and dead patients, we proceeded to modify the sentence according to reviewer’s suggestion, removing the confounding term “ICU discharge” with “last three days of ICU stay”. We have also proceeded to modify the sentence in the Conclusions, removing any terms that might appear promotion.
